# Cell Theranostics on Mesoporous Silicon Substrates

**DOI:** 10.3390/pharmaceutics12050481

**Published:** 2020-05-25

**Authors:** Maria Laura Coluccio, Valentina Onesto, Giovanni Marinaro, Mauro Dell’Apa, Stefania De Vitis, Alessandra Imbrogno, Luca Tirinato, Gerardo Perozziello, Enzo Di Fabrizio, Patrizio Candeloro, Natalia Malara, Francesco Gentile

**Affiliations:** 1Department of Experimental and Clinical Medicine, University Magna Graecia, 88100 Catanzaro, Italy; coluccio@unicz.it (M.L.C.); valentina.onesto@unicz.it (V.O.); stefania.devitis24@gmail.com (S.D.V.); tirinato@unicz.it (L.T.); gerardo.perozziello@unicz.it (G.P.); patrizio.candeloro@unicz.it (P.C.); 2Institute of Process Engineering, Technische Universität Dresden, 01069 Dresden, Germany; giovanni.marinaro@kaust.edu.sa; 3Institute of Fluid Dynamics, Helmholtz-Zentrum Dresden-Rossendorf (HZDR), 01328 Dresden, Germany; 4Department of Electrical Engineering and Information Technology, University Federico II, 80125 Naples, Italy; mauro.dellapa@gmail.com; 5Emerging Materials & Devices, Tyndall National Institute, Cork T12 R5CP, Ireland; alessandra.imbrogno@tyndall.ie; 6Department of Applied Science and Technology, Polytechnic University of Turin, 10129 Torino, Italy; enzo.difabrizio@polito.it

**Keywords:** nanoporous silicon, gold nanoparticles, drug delivery, cancer cells, theranostics

## Abstract

The adhesion, proliferation, and migration of cells over nanomaterials is regulated by a cascade of biochemical signals that originate at the interface of a cell with a substrate and propagate through the cytoplasm to the nucleus. The topography of the substrate plays a major role in this process. Cell adhesion molecules (CAMs) have a characteristic size of some nanometers and a range of action of some tens of nanometers. Controlling details of a surface at the nanoscale—the same dimensional over which CAMs operate—offers ways to govern the behavior of cells and create organoids or tissues with heretofore unattainable precision. Here, using electrochemical procedures, we generated mesoporous silicon surfaces with different values of pore size (PS ≈ 11 nm and PS ≈ 21 nm), roughness (Ra ≈ 7 nm and Ra ≈ 13 nm), and fractal dimension (Df ≈ 2.48 and Df ≈ 2.15). Using electroless deposition, we deposited over these substrates thin layers of gold nanoparticles. Resulting devices feature (i) nanoscale details for the stimulation and control of cell assembly, (ii) arrays of pores for drug loading/release, (iii) layers of nanostructured gold for the enhancement of the electromagnetic signal in Raman spectroscopy (SERS). We then used these devices as cell culturing substrates. Upon loading with the anti-tumor drug PtCl (O,O′-acac)(DMSO) we examined the rate of adhesion and growth of breast cancer MCF-7 cells under the coincidental effects of surface geometry and drug release. Using confocal imaging and SERS spectroscopy we determined the relative importance of nano-topography and delivery of therapeutics on cell growth—and how an unbalance between these competing agents can accelerate the development of tumor cells.

## 1. Introduction

Tissue engineering is a combination of techniques and materials for the fabrication of scaffolds and devices that, interacting with the cells, can lead to the formation of an analogue of tissues and organs that can improve, assist, or replace those already existing in the human body [1,2,3,4]. The biomaterials to be used in tissue engineering should exhibit the most convenient combination of mechanical properties, macro-scale architecture, and nanoscale geometry, to influence the collective behavior of cells and induce cells to form efficient structures. Those structures should be biocompatible, energetically efficient, autonomous, computationally efficient, and should be organized in a way to optimize the exchange of biochemical signals, nutrients, and oxygen between the several different parts of the structures and the external environment [1,2,4,5]. Thus, an ideal scaffold should present details over different hierarchical length scales to enable cell colonization, migration, and organization, and should be preferentially porous to enable the transport of bio-molecules.

At the nanoscale, cell behavior is strongly influenced by their interaction with the surrounding microenvironment [6,7,8,9,10,11]. Nanomaterials can interact without intermediation with the those adhesion molecules (integrins, cadherins, selectins, the immunoglobulin superfamily of cell surface proteins) involved in several different cell functions, including recognition, binding, adhesion, migration, apoptosis, differentiation, survival, and transcription [7,8,9,11,12]. Due to this unmediated interaction, nanomaterials can be fine-tuned to manipulate cellular function [13]. Materials with a controlled design at the nanoscale have been demonstrated in applications such as stem cells differentiation [14], the activation of the immune synapse [15], the shaping and signaling in neuronal networks [16], cell adhesion [17,18,19] and growth [20,21], the manipulation and control of neural polarity [22]. While many of the reported works have focused on bi-dimensional geometries, recent advances in additive manufacturing technologies [23], such as light assisted photopolymerization techniques [24], stereolithography [25], digital light projection, and two-photon polymerization [26], allowed a smooth transition from a 2D to a 3D design of the intended nano-structures [27]. The resulting cell culture models have an increased degree of complexity, an increased number of degrees of freedom, and exhibit a more faithful adherence to the 3D complex architectures of cells in living tissue and organs. This in turn enables to reproduce with an increased level of fidelity the natural microenvironment of cells [28].

Nevertheless, despite important advances in the production of scaffolds for tissue engineering applications, the development of characterization techniques has lagged behind the progress of fabrication. Recently, the demand for new devices for the analysis of the behavior of cells at the length scale of cell receptor is increasing. Those devices may reveal the fundamental biological mechanisms behind cell adhesion, migration, and organization at the cell-adhesion-molecule level, disclosing precious information for those interested in designing the scaffolds in the most efficient way.

In this paper, we present a mesoporous silicon device with details over multiple scales for cell culture, growth, and assembly. The device is functionalized with gold nanoparticle clusters that can be exploited to amplify the Raman signal measured at the cell interface. Thanks to the pores in the silicon matrix, the device can release drugs, growth factors, or other biomolecules to the cells over time. Thus, the device combines the ability of a scaffold to support cell growth with the ability of a drug delivery system to vehicle active-molecules to the cells adhering to the scaffold. The gold nanoparticles on the device enable to examine the combined effects of surface nano-topography and the delivery of drugs on cell adhesion and proliferation. In experiments in which we put in culture cancerous MCF-7 cells on the device, we measured the simultaneous effect of the pore size and of the delivery of an anti-tumor drug on the adhesive and proliferation properties of cells. Moreover, using Raman spectroscopy and a multivariate analysis of data, we mapped the spatial distribution of receptors expressed over the cell surface, and correlated that distribution to the nanoscale architecture of the device. We found that cells exhibit an increased ability to grow and to form clusters on substrates with smaller pore size (PS ≈ 11 nm) and roughness (R_a_ ≈ 7 nm), compared to substrates with larger pore size (PS ≈ 21 nm) and roughness (Ra ≈ 13 nm), in line with previous studies [29,30]. Both substrates deliver their payload efficiently, up to 10 days from the initial release, demonstrating high anti-cancer efficacy and killing up to 90% of cancerous cells on the smaller mesoporous substrate after 72 h from cell culture.

The multi-functional device that we developed can be used to evaluate the coincidental effects of (i) a timely administrated drug or nutrient and of the (ii) nanoscale characteristics of a surface on the efficacy of a therapeutic treatment, the functionalities of a scaffold, or a combination of the two. The device can be potentially used in applications that bridge traditional drug delivery, traditional tissue engineering and regenerative medicine, and diagnostics.

## 2. Methods

### 2.1. Fabrication of Mesoporous Silicon Surfaces

A detailed scheme of the fabrication of the Au-functionalized substrates is reported in Figure 1. Silicon substrates were electrochemically etched to obtain porous silicon. Porous silicon is a form of silicon with arrays of pores penetrating through its structure [31]. The average pore size (PS) determines the class of the porous silicon material: substrates with PS < 2 nm, 2 < PS < 50 nm, or PS > 50 nm are classified as nanoporous, mesoporous, macroporous silicon substrates, respectively [31]. In this work we produced mesoporous silicon substrates with two different non-overlapping values of pore size: MeP_1_ silicon substrates with PS_1_ ≈ 11 nm and MeP_2_ silicon substrates with PS_2_ ≈ 21 nm. We used P-type, 100 silicon wafers as a substrate. We cut the originating silicon wafers into regular square chips with a side of ≈1 cm. We then positioned the chips in an impermeable electrolytic cell where samples were exposed to a solution of hydrofluoric acid (HF) and ethanol or methanol, under the action of an external electric field [30]. As a result, hydrogen ions in solution were accelerated towards the silicon substrate etching the pores. Substrates with a different pore size were obtained by tuning the parameters of the technique: the intensity of etching current, the concentration of HF in solution, the type of neutral component in solution (ethanol or methanol), and the time of the process. MeP_1_ silicon with an average pore size of PS ≈ 11 nm was obtained using a mixture of HF, D.I. (de-ionized) water and ethanol in a proportion of 1:1:1 in volume. In the process, a value of current density of I = 20 mA/cm^2^ was applied for 5 min at 25 °C. MeP_2_ silicon with an average pore size of PS ≈ 21 nm was obtained using a mixture of HF, D.I. water, and methanol in a proportion of 5:3:2 in volume. In the process, a value of current density of I = 4 mA/cm^2^ was applied for 5 min at 25 °C. In all cases, the thickness of the porous layer is of some tens of micrometers. Since porous silicon is intrinsically hydrophobic [32], samples were oxidized in an oven at 200 °C for 2 h before use. The photoluminescence of mesoporous silicon was verified by imaging the light emission of the samples under UV radiation (365 nm).

### 2.2. Electroless Deposition of Gold Nanoparticles Clusters

Clusters of gold nanoparticles were deposited on the porous sample surface using electroless deposition techniques. In the technique, metal ions in solution are reduced on an autocatalytic surface to form solid deposits of that metal [33]. Following the methods reported in [34], we treated the porous silicon samples in a solution of HF and gold (III) chloride (AuCl_3_) in a concentration of 0.15 M (HF) and 1 mM (AuCl_3_) for 3 min at 50 °C. In solution, the ions of gold react with the exposed silicon surface yielding gold nanoparticles with an average particle size d ≈ 20 nm. Samples were then rinsed in D.I. water at room temperature for 30 s.

### 2.3. SEM Sample Characterization

SEM (Scanning Electron Microscopy) analysis was conducted with a Zeiss GeminiSEM 500 at Dresden Center for Nanoanalysis (DCN), TU Dresden, Germany. Two types of porous silicon samples were analyzed: mesoporous 1 (MeP_1_) and mesoporous 2 (MeP_2_). Both samples were provided with and without gold nanoparticles deposited on their surface. Samples were fixed on stubs with a long pin and then mounted on a carousel 9 × 9 mm sample holder. In order to fix the samples, a small amount of silver paint was applied between the edge of the silicon substrate and the stub. A further copper lever was screwed in order to secure the sample on the stub. Several images of the samples were acquired in High Vacuum mode at 3 kV, a magnification factor of 300,000, and a working distance of about 3 mm with an InLens Detector (ZEISS) for secondary electrons. In order to reduce the drift, a frame integration (N = 14) was performed. In this way, every frame was scanned and averaged 14 times.

### 2.4. AFM Sample Characterization

Sample nanotopography was verified using atomic force microscopy (ICON Atomic Force Microscope, Bruker, Coventry, UK). We measured the surface profile over a sampling area of 1 × 1 μm^2^, in a dynamic tapping mode in air. All measurements were performed at room temperature. During image acquisition, the scan rate was fixed as 0.5 Hz, while images were discretized in 1024 × 1024 points. We used Ultra-sharp Si probes (ACLA-SS, AppNano, Mountain View, CA, USA) with a nominal tip radius less than 5 nm to assure high resolution. Multiple measurements were done in different scan directions to avoid artefacts. At least four images were recorded per sample to reduce uncertainty. After acquisition, images were analyzed using the methods developed in [17] to determine the average surface roughness (Ra) and fractal dimension (Df) for each sample.

### 2.5. Contact Angle Characterization of Samples

The wettability of the samples was verified using an automatic contact angle meter (KSV CAM 101, KSV Instruments Ltd., Helsinki, Finland). A drop of 5 μL of D.I. water was gently positioned on the sample surface at room temperature. After 5 s from deposition, the contact angle of the drop at the interface with the substrate was measured.

### 2.6. MCF-7 Cell Culture and Staining

Breast carcinoma MCF-7 cells were grown on the porous silicon surfaces. Cultures were carried out at 37 °C in a humidified 5% CO_2_/air atmosphere in a Dulbecco’s modified eagle’s medium (DMEM, Euroclone) supplied with 10% heat-inactivated fetal bovine serum (Euroclone, Pero (Mi), Italy), streptomycin (0.2 mg/mL) and penicillin (200 IU/mL). When cells on the petri dishes reached 90% confluence, they were dissociated: medium was removed and MCF-7 were treated with a solution of 0.25% Trypsin-0.53mM EDTA (Euroclone) for about 5 min at 37 °C. Trypsin was deactivated by adding medium and completely removed after centrifugation of the cell suspension (1300 rpm, 5 min, 18 °C). Then, trypsin/growth medium solution was removed. Single sterilized porous Si wafer specimens with and without loaded drugs, having a size of around 15 × 15 mm, were individually placed into each well of a 6-well plate (Corning Incorporated) and washed with phosphate-buffered saline solution (PBS, Invitrogen). After that, cells were seeded in complete cell medium and cultured up to 15 days in a humidified incubator at 37 °C with 5% of CO_2_. After the incubation period, cell culture medium was removed and the MCF-7 cells were washed twice in PBS, fixed with 4% PFA (paraformaldehyde), and left for 30 min at room temperature (RT). Subsequently, cells were washed twice in PBS and permeabilized with 0.05% triton (Invitrogen, Milano, Italy) for 5 min at RT. Fixed and permeabilized cells were stained with 100 μL DAPI (40, 6-Diamidino-2-phenylindole, Sigma Aldrich, Milano, Italy) solution for 10 min at 4 °C in dark environment. In the end, the DAPI solute ion was removed and each sample was washed with PBS. The total number of cells n_tot_ initially seeded in each well for incubation was approximately n_tot_ ≈ 10^5^. Cells were sub-confluent for the duration of the experiment. MCF-7 cells were chosen as a cell-model because they are characterized by a moderate expression of the integrins. As previously reported [35], the change in the intensity and type of expression of integrin is the basis of the cancer disease progression. Notably, MCF-7 are a secondary cell line. The choice of another cell line, perhaps a primary cell line, while could possibly enhance clustering, may not have—at the same time—the same effect on the expression of integrins in the system.

### 2.7. Imaging Cells on the Substrates

An inverted Leica TCS-SP2^®^ laser scanning confocal microscopy (Wetzlar, Germany) system was used to image cells adhering on the substrates. All measurements were performed using ArUv laser (Leica, Wetzlar, Germany). The pinhole was set to ≈ 80 μm (1.5 Airy units) and the laser power to 80% of the maximum, these values of the parameters were maintained constant throughout each acquisition. Confocal images of blue (DAPI) fluorescence were acquired using a 405 nm excitation line and a 10× dry objective, so that several cells could be simultaneously imaged in the region of interest, that was of 1174 × 882 μm^2^, resulting in a pixel size of ≈ 1.72 μm. For each substrate, a large number of images was taken for statistical analysis. Each image was averaged over four lines and 10 frames to reduce noise. Images were acquired with a resolution of 1024 × 768 pixels, and were exported to a computer for processing and analysis.

### 2.8. Image Analysis and Topological Characteristics of Cell Networks

Confocal images of cell nuclei stained with DAPI were analyzed with Matlab^®^ to extract the cell positions. Images were preprocessed to enhance contrast and low-pass filtered to remove constant power additive noise. Each image was partitioned into k = k different segments (going gradually from bright, k = 1, to dark, k = k) by k-means segmentation algorithms [30]. The information content of the image was thus associated with a gray level k = t and all the segments brighter than a certain threshold t were considered as background and shifted to 0. The values of the remaining segments, representing the cells, were shifted to 1. After that, the resulting image (*g*) was downsampled: if *f* was the average operator, *f* was shifted over *g* by steps of size *r*, where *r*^2^ was the expected area of a cell nucleus in pixels. The pixel intensity (ranging from 0 to 1) of the resulting image indicated the probability that a pixel is a cell. If this probability is greater than a threshold, that pixel is considered being a node of the graph in a bi-dimensional grid. At this point, the links between the nodes can be established using the Waxman model [36], according to which the probability *P*(*u*,*v*) of being a connection between two nodes *u* and *v* exponentially decays with their Euclidean distance *d*. If *L* is the largest Euclidean distance:(1)$$P\left(u,v\right)=\alpha e^{-d\left(u,v\right)/\beta L}$$
where *d* is the Euclidean distance between nodes *u* and *v*, and *L* is the largest possible Euclidean distance between two nodes of the grid. In the equation, α and β are the Waxman model parameters. According to α and β, which have to be chosen between 0 and 1, the density of links in a graph changes. In particular, low values of these parameters result in a low number of connections. For our study, α = 1 and β = 0.025. The probability *P* ranges between 0 (if the distance between the pair of nodes is ideally infinite) and 1 (if the distance between the pair of nodes is zero). The connectivity information of a graph is described in the adjacency matrix A. A is a square matrix in which each element *a*_*ij*_ indicates whether two nodes *i* and *j* are connected (*a*_*ij*_ = 1) or not (*a*_*ij*_ = 0). In the analysis, diagonal elements were all zero, since links from a node to itself were not allowed. Moreover, graphs were considered being undirected, so that information could bidirectionally flow from *i* to *j*. As a consequence A was symmetric and *a*_*ij*_ = *a*_*ji*_. As the Euclidean distances *d*_*ij*_ in the networks were extracted, we could decide if a pair of nodes is connected by the subsequent formula
(2)$$\alpha e^{-d_{i,j}/\beta L}-R\geq 0$$
in which *R* is a constant that we chose as 0.1 so that the probability of being a connection is *P* = 0.9. Once established the connections between the nodes, the network parameters including *c*lustering coefficient, characteristic path length, and small-world-ness can be extracted. The definition and significance of these terms may be found in influential textbooks [37] and papers [38,39,40,41]. Once obtained the *Cc* and *Cpl* values, we found a precise measure of ‘small-world-ness’, the ‘small-world-ness’ coefficient (SW), based on the trade-off between high local clustering and short path length as described in [42]:(3)$$\mathrm{SW}=\frac{Cc_{graph}}{Cc_{rand}}/\frac{Cpl_{graph}}{Cpl_{rand}}$$
where *Cc*_graph_ and *Cpl*_graph_ are the clustering coefficient and the characteristic path length of the graph *G* under study, and *Cc*_rand_ and *Cpl*_rand_ are the equivalent values for a random Erdös-Rényi graph with the same number of nodes and edges of *G*.

### 2.9. Raman Analysis of Samples

MCF-7 cells fixed on the Au-mesoporous sample surface were analyzed by a WITec Raman microscope Alpha300 AR equipped with a 50×/0.7 N.A. (Numerical Aperture) objective. The signal was excited by a 633 nm laser, set to a power of 1 mW. For each sample, SERS (Surface Enhanced Raman Spectroscopy) maps of a portion of a cell were acquired in the *x*-*y* plane with a 0.5 μm stepsize, with the aim to analyze the SERS spectra coming in particular from the cell membrane, searching the different biochemical composition of each point to evidence the possible presence of adhesion proteins [43]. The Raman spectra were collected in the spectral range from 700 to 3250 1/cm, with an integration time of 1 s.

### 2.10. Principal Components Analysis of Raman Spectra

Spectra were pre-processed to minimize the effect of the fluorescence of samples by normalization on the total spectrum area. A principal component analysis (PCA) and a clustering analysis were performed on the spectral collection to highlight the chemical differences between the membrane’s points [44]. The first five principal components (PCs) accounted for nearly 90% of the total spectral variation and they were then used to implement the clustering analysis by the Kmean method, imposing a number of five classes. All the pre-processing steps, the PCA, and the clustering analysis were carried out using the free software package Raman Tool Set (available on http://ramantoolset.sourceforge.net).

### 2.11. UV Characterization of Drug Release

To assess the drug-delivery capability of the device we verified the release over time of the anti-tumor drug PtCl(O,O^-acac)(*DMSO*). PtCl(O,O′-acac)(*DMSO*) is a platinum(II) complex containing acetylacetonate (acac), characterized by a high toxicity both in immortalized cell lines, as human cervical carcinoma (HeLa) cells or human breast cancer (MCF-7) cells, and in primary cultured human breast epithelial cells [45]. In this work, we incubated the mesoporous silicon samples in 30 μM/50 μM solutions of PtCl(O,O′-acac)(DMSO) in D.I. water for 60 h to load the drug. The release kinetic was tested over time up to 15 days, by immersion of the loaded sample in DI water and monitoring the drug concentration at different time, through the analysis of drug in solution by a spectrophotometer UV/Vis (LAMBDA 25 UV/Vis PerkinElmer), after standard calibration procedures of the samples.

### 2.12. Statistical Analysis

Data in the article and in the figures are represented as mean ± standard deviation. We used a Student’s *t*-test statistics (two-tailed, unpaired) to perform comparison between means of different groups, where we assumed that elements in each group are normally distributed. In performing the test, the null hypothesis is that the means between pairs of samples are equal. Everywhere in the text and the figures the difference between two subsets of data is considered statistically significant if the Student’s *t*-test gives a significant level *p* (*p* value) less than 0.05.

## 3. Results

### 3.1. Producing Gold-Functionalized Mesoporous Surfaces

Using electrochemical etching techniques described in Section 2, we produced porous silicon surfaces. Tuning the parameters of the electrochemical etching, we obtained two different pore morphologies: (i) mesoporous silicon samples with a pore size that oscillates around the central value PS = 11 nm (MeP_1_ silicon with a pore size in the lower nanometer range) and (ii) mesoporous silicon samples with an average pore size PS = 21 nm (MeP_2_ silicon with a pore size in the higher nanometer range). Scanning electron micrographs (SEM) of MeP_1_ and MeP_2_ samples taken at different magnifications reveal the morphology of the porous surface at different scales (Figure 2a–f). Pores on the surface of MeP_1_ silicon are less uniformly distributed, are less dense, and result in a porosity of the sample of about P ≈ 12% (Figure 2a–c). The porosity or void fraction is a measure of the void spaces in a material, it is a fraction of the volume of voids over the total volume, expressed here as a percentage between 0% and 100%: it was calculated following the image analysis algorithms reported in the Appendix A. Differently, pores on the surface of MeP_2_ silicon are more uniform and are more densely packed compared to the pores found in the MeP_1_ morphology: for this category, the porosity of the sample soars to *P* ≈ 40% (Figure 2d–f).

Thus, the room potentially available to accommodate drugs or other molecules is significantly larger for MeP_2_ silicon than for MeP_1_ samples. After porosification, MeP_1_ and MeP_2_ samples were functionalized with gold nanoparticles using electroless deposition techniques. Electroless deposition is a technique that enables the reduction of metal ions on a solid surface as bulk metal without the application of external electric fields or forces [33]. Samples were treated with a solution of gold chloride and hydrofluoric acid for 3 min at 50 °C (Section 2). The process resulted in the homogeneous deposition of clusters of gold nanoparticles on the surface of the porous samples. SEM images of the sample surface (Figure 2d–f) and a convenient analysis of data (Appendix A) indicate that the average diameter of the gold nanoparticles is s_np_ = 8 nm, with a small deviation around the mean *σ*(s_np_) ≈ 1 nm. In no case do the gold nanoparticles occlude the pores, therefore, preventing the correct release of drugs of molecules from the porous matrix. Porous surfaces functionalized with gold nanoparticle were verified using atomic force microscopy (AFM). AFM imaging enabled to resolve the structure of the samples at the nanoparticle level for both MeP_1_ (Figure 3a) and MeP_2_ (Figure 3c) silicon. Fast Fourier transform of AFM data enabled to derive the power spectrum associated to MeP_1_ (Figure 3b) and MeP_2_ (Figure 3d) silicon functionalized with gold. The power spectrum reports the change of information content as a change of size in a bi-logarithmic scale thus indicating how much of the originating complexity is maintained by changing the degree of detail of a surface [46]. Upon analysis of AFM data we found the values of roughness (Ra) and fractal dimension (Df) of the samples as Ra^MeP1^ = 7 ± 2 nm and Df^MeP1^ = 2.48 ± 0.4 for MeP_1_, and Ra^MeP2^ = 13 ± 3 nm and Df^MeP2^ = 2.15 ± 0.2 for MeP_2_ silicon. Thus MeP_1_ samples exhibit values of roughness and fractal dimension larger than the corresponding values found for MeP_2_ silicon. For comparison, nominally flat silicon surfaces, used as a control, have significantly smaller values of roughness (Ra^Si^ = 1 ± 0.1 nm) and fractal dimension (Df^Si^ = 2.1 ± 0.2) (Figure 3h). The luminescence properties of porous silicon samples were verified under UV light (Figure 3e). The intense luminescence emission from MeP_1_ samples, compared to the low emission of MeP_2_ and to the no-emission from simple silicon, indicates that MeP_1_ samples may have—in the long tail of their pore size distribution—pores with a size smaller than 2 nm [31]. The wettability of mesoporous silicon samples was verified using contact angle measurements. Before oxidation, porous silicon samples as made exhibit a marked hydrophobicity with values of contact angles (CA) approaching 120° (Figure 3f). After treatment (Section 2), contact angle values measured on the sample surface shift to smaller values (CA ≈ 35°) typical of a hydrophilic surface.

### 3.2. Controlling Cell Organization on Au-Mesoporous Silicon Surfaces

The substrates that we produced exhibit different values of pore size (PS^MeP1^ ≈ 11 nm, PS^MeP2^ ≈ 21 nm), gold nanoparticles size (s_np_ ≈ 8 nm), roughness (Ra^MeP1^ ~ 7 nm, Ra^MeP2^ ≈ 13 nm), and fractal dimension (Df^MeP1^ ≈ 2.48, Df^MeP1^ ≈ 2.15). To examine whether the nano-topographical characteristics of the surfaces have the ability to direct cell behavior on the substrate in a controlled way, we put in culture on both Au-MeP_1_ and Au-MeP_2_ silicon MCF-7 breast cancer cells. We then examined the topological characteristics of the networks that cells formed 24 h from seeding and we correlated them to the topography of the surface. We used nominally flat silicon substrates as a control. Figure 4 shows the spatial layout of cell-nuclei on flat silicon, Au-MeP_1_ and Au-MeP_2_ silicon imaged 24 h after culture. The initial number of cells deposited in each well for incubation was the same for all the substrates (Section 2). Fluorescence images in Figure 4 show that cells are homogeneously distributed on flat silicon surfaces, showing no preferential points of accumulation. Differently, cells on mesoporous surfaces form complex structures of those cells with a correlation length, cluster size, and topological characteristics that seem to vary from Au-MeP_1_ to Au-MeP_2_ silicon.

We used image analysis algorithms and the methods of networks analysis, described in [30,40,47,48] and Section 2 of this article, to measure the characteristics of cell-networks quantitatively. Starting from the fluorescence image of a cell configuration, we segmented that image to find the cell-centers. Then, we routed cell-centers using the Waxman algorithm (Figure 5a). The algorithm selected the cell pairs to be connected basing on their distance: cells that were closer than a threshold were connected as described in Section 2. We analyzed more than 30 images per substrate. For each image, we extracted from the resulting network the number of cells in a region of interest (*N*), the clustering coefficient (*Cc*), the characteristic path length (*Cpl*), the small-world-ness (SW). N measures the adhesion strength of cells to a substrate [17]. The clustering coefficient, characteristic path length, and small world coefficient are a quantitative measure of the characteristics of the networks that cells form on a surface. The clustering coefficient is the ratio of active links to the total combinations of connections that cells, around a reference node, can possibly establish, averaged all over the cells of the network [37]. The characteristic path length is the mean shortest distance between nodes of network [37]. The small-world coefficient, found as a combination of the *Cc* and the *Cpl* [42], is a metric that tests whether the distance between nodes grows with the logarithm of the number of nodes in a graph: *Cpl*∝log(*N*). Typically, small-world-networks are characterized by a few clusters with a high number of elements for a cluster, the identification of small-world networks is of interest because networks with small-world-characteristics communicate more efficiently than equivalent random or ordered graphs of the same size [48,49].

After network analysis of cell images, we found that the number of adhering cells in a region of interest of 1174 × 882 μm is *N* ≈ 663 on flat silicon, *N* ≈ 884 for the Au-MeP_1_ substrate, *N* ≈ 544 for the Au-MeP_2_ substrate (Figure 5b). The maximum number of adhering cells is found for the Au-MeP_1_ substrate with intermediate values of roughness and higher values of fractal dimension, in line with previous reports [17,29,30,48,49]. Notably, the difference between the number of cells found on Au-MeP_1_ and Au-MeP_2_ silicon is statistically significant (*p* < 0.05). For the same sets of images, we found that the values of clustering coefficient reach a maximum for the Au-MeP_1_ substrate, with *Cc* ≈ 0.74, while the clustering coefficient is lower for the Au-MeP_2_ substrate (*Cc* ≈ 0.65), and significantly lower for simple silicon (*Cc* ≈ 0.58) (Figure 5c). At the same time, the values of characteristic path length are nearly identical for the Au-MeP_1_ (*Cpl* ≈ 2.43) and Au-MeP_2_ (*Cpl* ≈ 2.15) substrates, and they are statistically different from the values found on flat silicon with *Cpl* ≈ 8 (Figure 5d). The resulting small-world-coefficient of cell networks on mesoporous substrates is SW ≈ 1.29 and SW ≈ 1.35 for MeP_1_ and MeP_2_, respectively, while SW ≈ 0.35 for flat silicon. Thus, cell networks on nanostructured, mesoporous surfaces passed the small-world test, differently from cells on flat silicon that settle on a surface without any appreciable large- or small-scale structure.

### 3.3. SERS Analysis of Cell Adhesion on Au-Mesoporous Silicon Surfaces

The substrates that we produced induce cell clustering. The increased susceptibility of cells to condensate into compact structures is in turn ascribed to the intermediate values of roughness and large values of fractal dimension of mesoporous silicon compared to flat silicon substrates [17,30]. Since cell clustering and condensation is a side effect of the of the increased adhesive properties of a substrate [48,49], we performed a chemometric analysis of cells cultured on MeP_1_ and MeP_2_ substrates to examine whether cell adhesion molecules are preferentially expressed from cells on nanostructured surfaces. We mapped the Raman intensity of MCF-7 cells cultivated on MeP_1_ and MeP_2_ silicon functionalized with gold, compared to the same cells on silicon sputtered with a continuous layer of gold, used as a control (Figure 6a). Raman spectra of cells were acquired following the procedure reported in Section 2 36 h after seeding, that is a sufficiently long time to assure complete adhesion of cells on the substrate. In each point, the Raman maps reported in Figure 6a are proportional to the intensity of the spectra measured at 1569 1/cm, that is typical of integrins as explained in the following of this section. Raman spectra were then subjected to a principal components analysis (PCA), and the principal components resulting from the analysis were in turn clustered into groups using classical k-means algorithms. This allowed to identify in the cell under analysis of five different regions, where points in a region have similar chemometric characteristics (Figure 6b). Moreover, since the principal components are sorted in order of decreasing information content and variance [50], regions in Figure 6b define the portions of the cell that exhibit the more intense and the more vibrantly varying Raman signal.

Integrins are one of four principal cell adhesion molecules families, they play a major role in the process of adhesion of cells to the extracellular matrix (ECM), and especially in tumor cells where they are overexpressed during the process of adhesion [51]. α5β1 and α3β1 are integrins specifically expressed by tumour and epithelial cells. In particular, α3β1 is overexpressed in tumours spreading in ECM with a high content of collagen and laminin, so that an elevated concentration of α3β1 is a hallmark of cell proliferation and migration [52]. While each of those adhesion molecules have their own distinctive features, nonetheless they exhibit a certain number of peaks that do not vary from spectrum to spectrum representing a fingerprint for those molecules. Those peaks are found at (i) 1126 1/cm related to C-N bond, (ii) 1175 1/cm associated to Tyrosine or Phenylalanine, (iii) 1306 1/cm attributable to amide III, (iv) 1506 1/cm related to Phenylalanine or Hystidine, (v) 1569 1/cm originating from tryptophan, (vi) 1645 1/cm due to the amide I signal [53]. The Raman analysis that we performed on cells on different surfaces was enhanced by the interaction of the electromagnetic (EM) field with gold nanostructures, which amplify the Raman signal by several orders of magnitude in a SERS (surface enhanced Raman spectroscopy) effect [54]. SERS analysis of cells enabled the identification of adhesion markers that are otherwise inaccessible to classical spectroscopy techniques. The Raman intensity profile in Figure 6a for MeP_1_ and MeP_2_ follows a characteristic and distinguishable spatial distribution. Points in the map with higher values of Raman intensity may be indicative of the expression of integrin cell-adhesion-molecules suggesting that in those spots adhesion is established. The map relative to simple silicon with gold shows less preferential points of adhesion, indicating that smoother unstructured surfaces impair cell adhesion and proliferation. Moreover, SERS maps of cells measured on ${\mathrm{MeP}}_{1}$ and ${\mathrm{MeP}}_{2}$ substrates show a very high correspondence to the principal components distribution in Figure 6b, and especially to the first two components ${\mathrm{PC}}_{1}$ and ${\mathrm{PC}}_{2}$. Regions in the maps with greater overlap are at the borders of the cells (Figure 6a,b), with their membrane actively involved in the process of adhesion. The diagram in Figure 6c reports the loading associated to the first principal component measured for the ${\mathrm{MeP}}_{1}$, MeP_2_, and Si substrates. The loading is a statistical measure of how much different frequencies contribute to a certain principal component. The curves in Figure 6c indicate that the frequency that above all is responsible for the signal is 1569 1/cm for the MeP_1_ substrate, while the signal content associated to 1569 1/cm is gradually weaker for the ${\mathrm{MeP}}_{2}$ and the simple silicon substrate. Recalling that 1569 1/cm is the distinctive frequency for the integrins, the form of the diagrams of Figure 6c suggests that cell adhesion molecules are preferentially expressed for the foremost on MeP_1_ substrates, followed by MeP_2_ substrates and by flat silicon. For each substrate we calculated the ratio *r* between the value of loading intensity measured at 1569 1/cm and the loading averaged over the entire spectral range (Figure 6d). *r* is a quantitative measure of the relative abundance of integrins at the interface of a cell with a surface. Values of *r* much larger than one for MeP_1_ (*r* ≈ 4.53) and MeP_2_ (*r* ≈ 4.34), compared to the lower value of *r* measured on silicon (*r* ≈ 1), show that the multiscale architecture of mesoporous substrates, with meso-pores functionalized with metal nanoparticles, facilitate cell adhesion compared to flat geometries. The values of *r* determined for the MeP_1_ and MeP_2_ substrates are significantly different from that determined for silicon, with *p* < 0.05.

### 3.4. Kinetics of Drug Release from the Mesoporous Silicon Matrices

The devices that we produced incorporate networks of nano-pores penetrating deep within their structures. We verified the capability of the device to accumulate and consequently release drug molecules over time using UV spectroscopy techniques as described in Section 2. We incubated MeP_1_ and MeP_2_ silicon substrates with the antitumor drug PtCl(O,O′-acac)(DMSO) for 60 h. We used two different concentrations of the originating drug during the loading process: *c*_1_ = 30 μM and *c*_2_ = 50 μM. We then measured the release of the drug in D.I. water up to 10 days from the activation of the process. The diagram in Figure 7a shows the cumulative dose–response curves for different substrates and different initial values of the loading concentration. The dynamics of release from the MeP_2_ silicon system is faster compared to MeP_1_ silicon, consistently with the fact the pores in ${\mathrm{MeP}}_{2}$ silicon are larger (≈21 nm) than those contained in MeP_1_ silicon (≈11 nm). Wanting to approximate the curves of release with a function of time of the form $c\left(t\right)=c_{o}+c_{s}\left(1-e^{-t/\tau }\right)$, we found after nonlinear fitting of data the following solutions for c_s_ and τ: (i) *c*_s_ ≈ 4.94 μM and τ ≈ 46 h for MeP_1_ silicon loaded with an initial concentration equal to *c*_1_, (ii) *c*_s_ ≈ 2.81 μM and τ ≈ 50 h for MeP_1_ silicon with an initial *c*_2_ concentration, (iii) *c*_s_ ≈ 6.7 μM and τ ≈ 38 h for MeP_2_ silicon with an initial *c*_1_ concentration, *c*_s_ ≈ 7.2 μM and τ ≈ 20 h for MeP_2_ silicon with an initial $c_{2}$ concentration. *c*_s_ is the steady state value of the concentration increment with respect to a zero reference value. τ is the time constant of the drug delivery system, i.e., the time necessary to the system to reach 66% of its final value of concentration. The values that we found for c_s_ and τ for the different combinations of substrate morphology and initial loading concentration that we used in our study, indicate that the rate of drug release (1/τ) increases moving from MeP_1_ to MeP_2_ and, for the same substrate, it is higher for an initial higher concentration of the loaded drug. This behavior can be easily described by the first law of Fick, J=−D∂c/∂x, where the intensity of the flux (J) is proportional to the gradient of concentration from the substrate to the external environment, and the absolute number of molecules transported through the system per unit time depends on the area of the surface actively releasing the drug. In the equation D is the molecular diffusion coefficient [50]. Moreover, the values of *c*_s_ and τ and Figure 7a indicate that the total amount of drug released in a system (*c*_s_) is higher for MeP_2_ silicon, that has a higher porosity compared to MeP_1_, and is higher for an initial higher concentration of the loaded drug. In the neighbor of *t* = 0, the drug release profile can be expanded in a Taylor series yielding, neglecting higher order terms of the expansion, *c*(*t*) ≈ *c*_s_⁄τ: this approximate formula enables calculation of the velocity of release (*v*) at the early stage of the delivery process. Using data from the model fit, we obtained *v* ≈ 0.06 μM/h (MeP_1_, *c*_1_), *v* ≈ 0.11 μM/h (MeP_1_, *c*_2_), *v* ~ 0.17 μM/h (MeP_2_, *c*_1_), v ≈ 0.37 μM/h (MeP_2_, *c*_2_). Thus, the kinetics of initial release can be varied in the 0.06-0.37 μM/h interval by changing the parameters of the process. Figure 7b displays the drug released from the systems over time, normalized to the initial concentration of the loaded drug. Values in the figure are a measure of the efficiency of the drug delivery system. Data show that the maximum efficiency of release varies between ≈0.12 for MeP_1_ silicon with an initial loading concentration *c*_2_, and ≈0.23 for MeP_2_ silicon with an initial loading concentration *c*_1_. While the efficiency of delivery is still larger for MeP_2_ silicon with higher values of pore size and porosity, it decreases for increasing values of initial loading concentration, possibly because for larger amounts of initial payload the losses associated to the process are also larger. Thus, one of the points of strength of this bio-chip, is that it can artificially increase the half-life of a drug. The half-life (*t*½) is the time required to change the amount of a drug in the body by one-half during elimination. Drug clearance from the body is the result of elimination by renal excretion and by non-renal pathways, the latter most often represent clearance by the liver. Remarkably, the characteristic half-life—the duration of action—of anticancer drugs is, on average, small. Clinical studies and reports [55] indicate that the mean half-life of more than 140 small-molecule drugs approved for oncology indications is ≈15 h, with an even smaller value of median of about ≈5 h. The drug delivery system set-up in this study enables the active release of drugs for more than ≈50 h, depending on the configuration. Thus, the chip that we produced can possibly increase the half-life of most anticancer treatments by 300% on average.

### 3.5. Timely Delivery of Drugs Impairs Tumor Cells Adhesion

The theranostics device presented in the work has the ability to deliver over time the antitumor drug PtCl (O,O′-acac) (DMSO) for several hours from the initial release. We examined the effects of a controlled release of drugs on the adhesion and growth of tumor MCF-7 cells on the mesoporous substrates. MCF-7 cells were cultured on MeP_1_ silicon and MeP_2_ silicon functionalized with gold nanoparticles and on flat silicon surfaces used as a control. Half of the mesoporous substrates used in this study were loaded with PtCl (O,O′-acac) (DMSO) in a concentration of 50 μM. We then imaged cell-nuclei on different substrates and at different times from incubation. The different number of cells adhering on the substrates is the effect of a combination of factors: (i) the drug released from the system and (ii) the different nano-topographical characteristics of the substrates. Visual examination of samples reveals that already 6 h from culture the number of adhering cells on flat silicon is lower than that observed on MeP_1_ with gold, that is in turn different from the number of cells on MeP_1_ silicon loaded with drugs (Figure 8). This difference is exacerbated by time. In Figure 9a we report the bar-chart of the number of cells (*N*) measured on different substrates 36, 48, and 72 h from incubation. *N* was estimated from more than 20 images per substrate and five technical repeats per sample.

A total of 36 h after culture, the number of cells on MeP_1_ and MeP_2_ silicon without drug is not statistically different from that measured on flat silicon, with N_MeP1_ ≈ 1871, *N*_MeP2_ ≈ 2100, and *N*_Si_ ≈ 1951. Diversely, *N* is significantly lower for the mesoporous substrates loaded with drug, being *N*_MeP1_^drug^ ≈ 675 and *N*_MeP2_^drug^ ≈ 424. A total of 36 h from incubation the effect of topography is negligible compared to the release of drug.

A total of 48 h from incubation, the number of cells on MeP_1_ silicon without drug increases to *N*_MeP1_ ≈ 2928, significantly larger than the number of cells measured at the same time on simple MeP_2_ and flat silicon, with *N*_MeP2_ ≈ 1983 and *N*_Si_ ≈ 2090. At this time step, the number of cells measured on the mesoporous substrates loaded with drug falls to *N*_MeP1_^drug^ ≈ 451 and *N*_MeP2_^drug^ ≈ 255, that are significantly different from the values measured for the unloaded substrates. A total of 48 h after cell culture, the improved adhesive characteristics of MeP_1_ silicon over simple MeP_2_ and simple silicon become apparent, while the delivery of drugs from MeP_2_ silicon achieves maximum effects.

A total of 72 h from incubation, the number of adhering cells on MeP_1_ silicon with drug hits a minimum, *N*_MeP1_^drug^ ≈ 362, while *N* surges to *N*_MeP1_ ≈ 3962 for simple MeP_1_ silicon without drug. These values are significantly lower and significantly larger than the control: *N*_Si_ ≈ 2697. At this time of the analysis, the MeP_2_ morphology does not enhance significantly adhesion with respect to the control, with *N*_MeP2_ ≈ 2462, while the release of drug from MeP_2_ silicon still bears appreciable effects, with N_MeP2_^drug^ ≈ 1355 significantly lower than the number of cells measured on flat silicon.

Thus, the efficacy of the drug delivery system depends on both of the substrate morphology and the time of the process. For each time, we calculated the efficiency of delivery as the number of adhering cells on the substrate loaded with drug, to the number of cells measured on the same substrate without drug (Figure 9b,c). We observe that the efficiency of the system gradually increases for MeP_1_ silicon, varying from ≈0.64 at 36 h, to ≈0.84 at 48 h, to ≈0.90 at 72 h from incubation. For this substrate, more than 90% of cells are killed compared to the same substrate in absence of drug delivery. The curve of efficiency for the MeP_2_ silicon is different. For this system, the efficiency of delivery is sufficiently large already 36 h from the initial release (≈0.80), it attains a maximum value at 48 h (≈0.87), to subsequently drop to ≈0.45 at 72 h from incubation. The different efficiency measured for MeP_1_ and MeP_2_ silicon reflects the different kinetics of release measured for those devices and reported in previous sections of this work.

## 4. Discussion

The theranostics substrates that we produced are akin to the cell culture glass coverslips used to plate cells for research and analysis—with the notable exception that they enable the targeted delivery of therapeutics to the cells and cell sensing, simultaneously. Moreover, thanks to the fabrication process capability to attain maximum control over surface morphology at the nanoscale, the substrates can be designed to guide cell adhesion, proliferation, and organization. For these characteristics, this bio-device—and its more sophisticated evolutions that will be developed over time—can be integrated into conventional cell culture dishes or multi-wells to test the adhesion and growth of cells against different external factors, including substrate geometry and a controlled delivery of drugs. Researchers can plate cells over several different replica of the device, each of them with its characteristic topography and drug release profile. Then, the researchers will find the combination of surface topography and device payload that guarantees maximum/minimum cell adhesion and proliferation, depending on whether the aim of the research is optimize a structure for tissue engineering or the effects of a drug for personalized medicine. The search for the optimal values of surface topography and kinetics of release should be possibly conducted within the bounds identified by this and other similar works: where the roughness of the surface and the pore size is varied in the 0–30 nm interval, while drugs are released with a maximum initial rate of ≈0.4 μM/h. The output of the experiment—i.e., cell colonies—can be verified at different time steps from seeding using either confocal microscopy or Raman spectroscopy that is, notably, made possible by the distinctive design of the device. While the first technique provides information about cell adhesion, growth, and clustering, Raman analysis of samples describes the conditions of a cell at the level of its adhesion molecules. Thus, the combination of techniques gives a picture of the evolution of a cell over different scales, bridging the divide between the behavior of cells being observed in isolation (individual behavior of cells) or in-group (collective behavior of cells). Consistency between results may indicate that the substrate operates efficiently towards either improving or impairing cell adhesion and organization. Thus, the device can potentially be the basis for a test campaign aimed to optimize the characteristics of biomaterials for tissue engineering, regenerative medicine, or in-vitro-model applications. After identification of the optimal surface characteristics and drug dosage that assure the wanted effect, these values should be copied to the scaffold intended to support cell-growth, or to the implantable device that will release drugs to a disease, for real-life applications. Nonetheless, this implies a process of engineering of the device, aimed to overtake those complications that can possibly emerge when similar devices are used outside of a research context. A list of possible caveats is identified as follows:(1)Transition from a 2D to 3D geometry. Results of the work and the way the device operates are restricted to 2D geometries. Cells themselves are plated on a surface and cell clusters are described using bi-dimensional variables. Future research that will be conducted over time shall have to clarify whether, and to which extent, cell behavior changes moving from 2D to 3D scaffolds.(2)Understanding whether the delivery is active. As demonstrated in the work, the delivery of the drugs from the porous matrix does not last indefinitely. Additionally, there could be cases in which, because of unpredictable leakages or occlusions of the pores, the process of release terminates before the expected time. Thus, the substrate should incorporate a sensor sensitive to the released drug, indicating the rate at which release proceeds, warning of possible malfunctions of the device before cells react to the alterations of delivery.(3)Switching between on-off delivery states of the device. In this configuration, the release of drugs is always active, being driven by the gradients of concentration in the system—and described by the Fick’s laws. In a more sophisticated evolution of the device, one should be able to switch between on-off states: i.e., an active (on) state, in which drug molecules are allowed to flow in the system, and an inactive (off) state, in which release is temporarily paused. This can be possibly accomplished by varying, in a controlled fashion, the levels of pH and temperature of the system, having previously conjugated the drug with a pH-sensitive cleavable linker as described, as for an example, for injectable nanoparticle generators in [56].(4)Increasing the efficiency of drug loading, either in terms of pore-capacity and total amount of drug loaded in the pores, and in terms of loading time. The time necessary to load the drug into the device (more than 60 h for the present configuration) is unacceptably long for real life/industrial applications of the device.(5)Integration. While the device offers complementary ways to measure cell performance, using the bio-chip alternatively in a confocal-microscope or in a Raman setup can take time, and slow down the pace of a test campaign, a trial, or a biology/medicine application of the device. The method can take advantage from the integration of the chip in an automatic multi-well plate reader, where different imaging techniques (confocal imaging and Raman spectroscopy) are combined in the same platform.(6)Optimization. The device can be especially useful in cancer theranostics. While the results obtained in this work are based on an initial cell density of about 10^5^ cells per substrate, this value can be optimized: the total number of cells for which the device has some measurable effects may be significantly lower than the 10^5^ limit. Miniaturization has as principal consequence the possibility to use the devices in all those cases in which, because of the early stage of a disease, conventional biopsies are ineffective. For the same reason, the device can be used to perform liquid biopsy. Liquid biopsy is based on the detection and isolation of cancer cells directly from the peripheral blood of the patient [57]. The limit of the method is that, often, cancer cells are too few to be detected. Liquid biopsy could realistically benefit from multi-drug array panels based on mesoporous silicon substrates, designed to evaluate the sensitivity of circulating tumor cells to a test drug.(7)Generality. Further to the end of liquid biopsy: results of the paper suggest that this theranostics device is also effective towards more aggressive cancer cells, such as triple negative breast cancer cells. The integrin expression that we chose to analyze in this work is believed to be a hallmark of more aggressive forms of breast cancer [58]. In particular, the expression of β1 integrins on the cell surface is a predictive marker of triple negative breast cancer. Since the substrates that we produced induce, for certain configurations, an increased production of integrins on the MCF-7 surface, it is legitimate to hypothesize that this theranostics procedure is effective also in the case of triple negative breast cancer cells. Regarding the impact on normal cells, it has been demonstrated in several studies—some of which have been cited throughout the article—that both the porosity and roughness of nanoscale surfaces affect the adhesion, proliferation, and clustering of cells, depending on their degree of differentiation and replication speed. Thus, the method that we developed is general in scope and can be realistically adapted to several different cell types for different applications.

## 5. Conclusions

Using electrochemical etching, we produced mesoporous silicon substrates functionalized with gold nanoparticles. Due to their porous structures, the substrates behave as a drug delivery system, where drugs or other agents loaded in the matrix are released over time with a first order kinetics and a time constant that can be varied by tuning the characteristics of the pores. Due to its nanostructure, the device can amplify by several orders of magnitude the signal generated by molecules in the cell membrane during the process of adhesion and migration. These nanodevices combine therapy and diagnostics effects in the same device, with the primary advantage of not being limited to therapy or sensing, as they provide coincident diagnostic information plus delivery of therapeutics. For their capabilities to drive cell growth and high capacities of therapeutic loading, these devices can be possibly used in tissue engineering, regenerative medicine, and nanomedicine. In these fields, the ability to assemble cells together has to be associated with the ability of deliver therapeutics accurately to achieve maximum control over cell fate. The ability of the system to measure adhesion at the nanoscale, is an extra feature that makes this theranostics device the ideal candidate for those who want to design substrates for cell growth and proliferation or, vice versa, induce apoptosis in tumor cells with unprecedented control.

In this work, by varying the parameters of the process we obtained substrates with different values of pore size, porosity, and roughness. MeP_1_ substrates exhibit smaller values of pore size (PS ≈ 11 nm) and roughness (Ra ≈ 7 nm), but have a larger fractal dimension (Df ≈ 2.48) compared to MeP_2_ substrates, with PS ≈ 21 nm, R_a_ ≈ 13 nm, and Df ≈ 2.15. Adhesion of MCF-7 cells was accelerated on MeP_1_ substrates with larger value of fractal dimension compared to MeP_2_ substrates and flat silicon used as a control.

When we considered the effect of the release over time of an anti-tumor drug, we observed that the maximum reduction of cell growth is found for MeP_2_ substrates 36 and 48 h after seeding, with a decrease in the number of adhering cells up to 87% with respect to the same substrates without drug. A total of 72 h from cell culture, the therapeutics efficacy of MeP_2_ substrates falls to 44%, and is overtaken by MeP_1_ silicon, with a cell number contraction of 90%, reflecting the different morphological characteristics of surfaces.

Results suggest that the adhesive properties of mesoporous substrates, the kinetics of drug delivery, and the effects that a combination of the two may have on the adhesion and proliferation of cells, can be conveniently modulated by changing the pore size and roughness in the narrow intervals of PS ≈ 11–21 nm and Ra ≈ 7–13 nm.

## Figures and Tables

**Figure 1 pharmaceutics-12-00481-f001:**
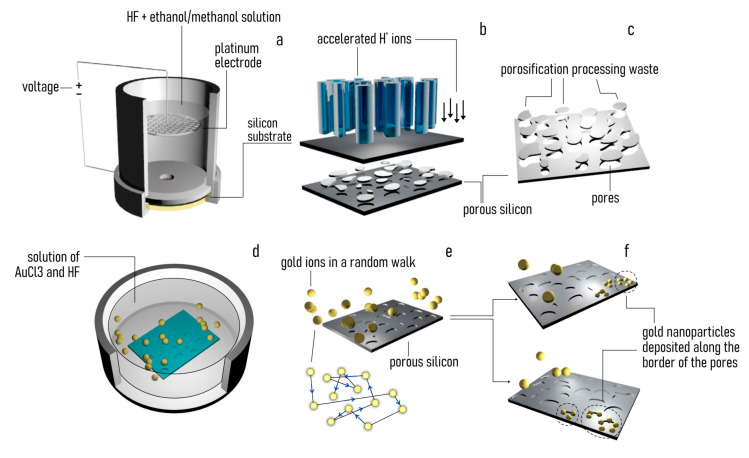
(**a**) An initial silicon chip of approximately 1 × 1 cm is electrochemical etched using a Teflon cell containing a solution of hydrofluoric acid (HF), D.I. (de-ionized) water, and methanol/ethanol in different ratios. (**b**) Upon activation of an external controlled voltage, positive ions in solution are accelerated towards the silicon substrate, creating pores within its structure. (**c**) Depending on the parameters of the process, including etching time, current and voltage intensity, and the concentration of the reagents in solution, one can obtain porous silicon surfaces with a tailored morphology. (**d**) The porous silicon sample is then placed in a baker along with a solution of hydrofluoric acid (HF) and gold(III) chloride (${\mathrm{AuCl}}_{3}$). (**e**) The resulting electroless process enables the deposition of gold ions in solution on the autocatalytic porous-silicon surface. (**f**) By varying the parameters of the electroless process, including temperature, concentration, and time, one can produce substrates with controlled gold-nanoparticles shape, size, and density.

**Figure 2 pharmaceutics-12-00481-f002:**
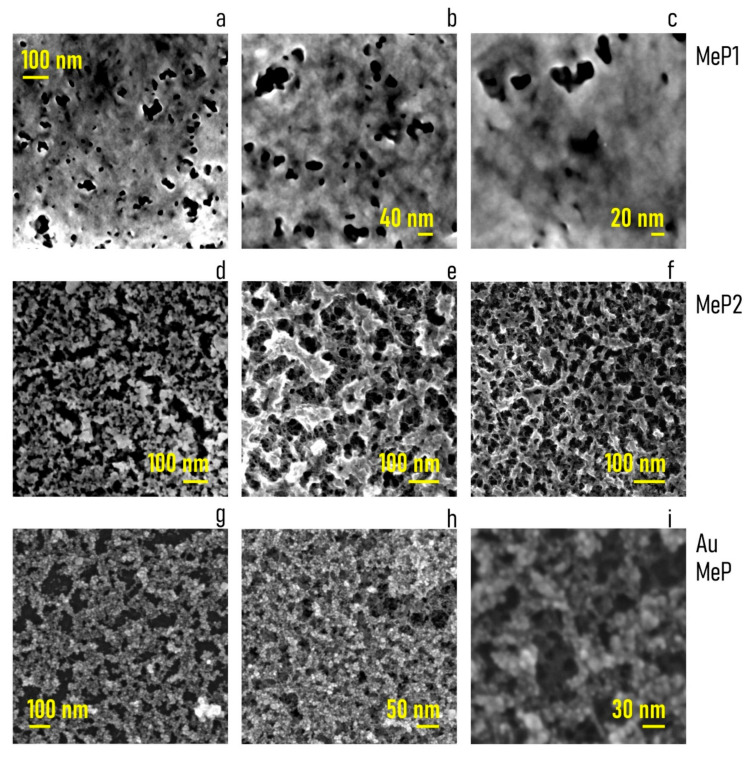
(**a**–**c**) SEM micrographs of MeP_1_ substrates at different levels of magnification, the pores are less uniformly distributed over the sample surface with an average pore size of PS ≈ 11 nm. (**d**–**f**) SEM micrographs of MeP_2_ substrates at different levels of magnification, the pores are uniformly distributed over the sample surface with an average pore size of PS ≈ 21 nm. (**g**–**i**) SEM micrographs of clusters of gold nanoparticles deposited over the porous sample surfaces, the particles follow the profile of the samples without occluding the pores, the average particle size is ≈8 nm.

**Figure 3 pharmaceutics-12-00481-f003:**
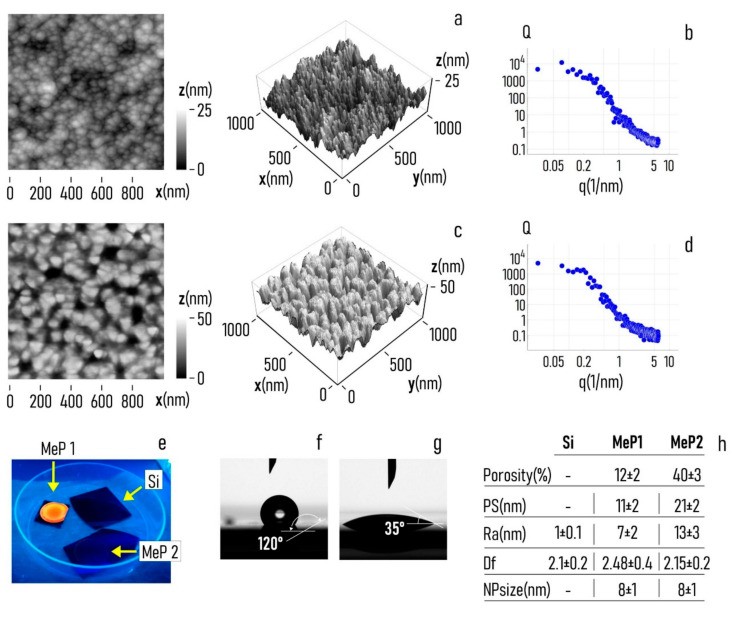
(**a**) Atomic force microscopy (AFM) profile of MeP_1_ substrates functionalized with gold nanoparticles imaged over a sampling area of 1 × 1 μm, the height values of the profile fall within the 0–25 nm range. (**b**) Power spectrum density function associated to the topography of the MeP_1_ substrate, the slope of the function in the linear regime is indicative of the fractal dimension of the samples. (**c**) AFM profile of MeP_2_ substrates functionalized with gold nanoparticles imaged over a sampling area of 1 × 1 μm, the height values of the profile fall within the 0–50 nm range. (**d**) Power spectrum density function associated to the topography of the MeP_2_ substrate. (**e**) Luminescence of MeP_1_ and MeP_2_ samples under UV light, compared to the light emission of silicon. Contact angle of a drop of D.I. water measured on the porous surfaces before (**f**) and after (**g**) sample oxidation. (**h**) Values of porosity, pore size, roughness, fractal dimension, and characteristic size of the gold nanoparticles of the porous surfaces determined through analysis of SEM and AFM images of samples.

**Figure 4 pharmaceutics-12-00481-f004:**
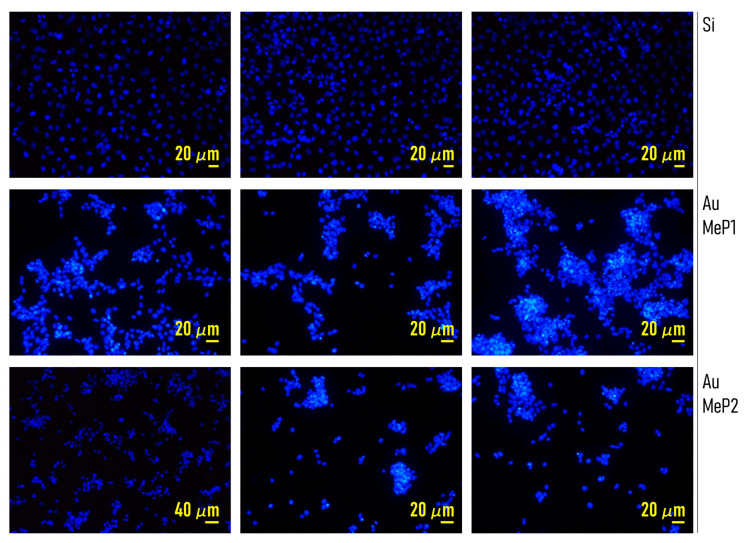
Fluorescence images of MCF-7 cancer cells over MeP_1_, MeP_2_, and silicon surfaces after 24 h from seeding.

**Figure 5 pharmaceutics-12-00481-f005:**
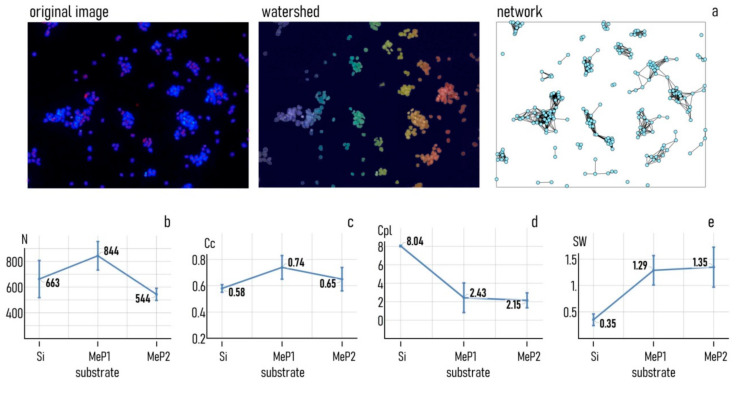
(**a**) Image analysis of fluorescence images: the original images were segmented with a watershed algorithm to identify individual cells, and cell nodes were then connected using the Waxman algorithm to obtain the equivalent graph for each sample. (**b**) Values of adhering cells, (**c**) clustering coefficient, (**d**) characteristic path length, and (**e**) small-world-ness determined for the cultures of MCF-7 cell on ${\mathrm{MeP}}_{1}$, ${\mathrm{MeP}}_{2}$, and silicon surfaces 24 h from seeding.

**Figure 6 pharmaceutics-12-00481-f006:**
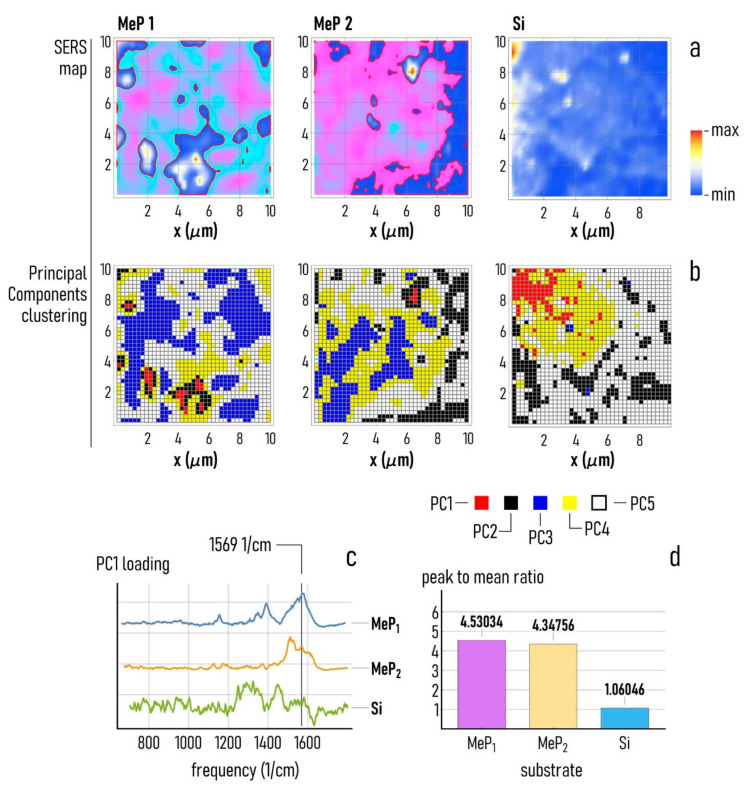
(**a**) Raman maps of MCF-7 cells acquired over a region of interest of 10 × 10 μm for MeP_1_, MeP_2_, and silicon surfaces, the maps show the Raman intensity measured at 1569 1/cm. (**b**) We show, for each of the considered surfaces (MeP_1_, MeP_2_, and Si), the first five principal components extracted from the Raman maps, and the spatial distribution of the principal components correlate with the Raman intensity maps previously reported. (**c**) The loading associated to the first principal component measured over MeP_1_, MeP_2_, and Si surfaces. (**d**) Ratio between the maximum and the mean intensity of the PC1 loading correspondent to MCF-7 cells cultivated over MeP_1_, MeP_2_, and Si samples.

**Figure 7 pharmaceutics-12-00481-f007:**
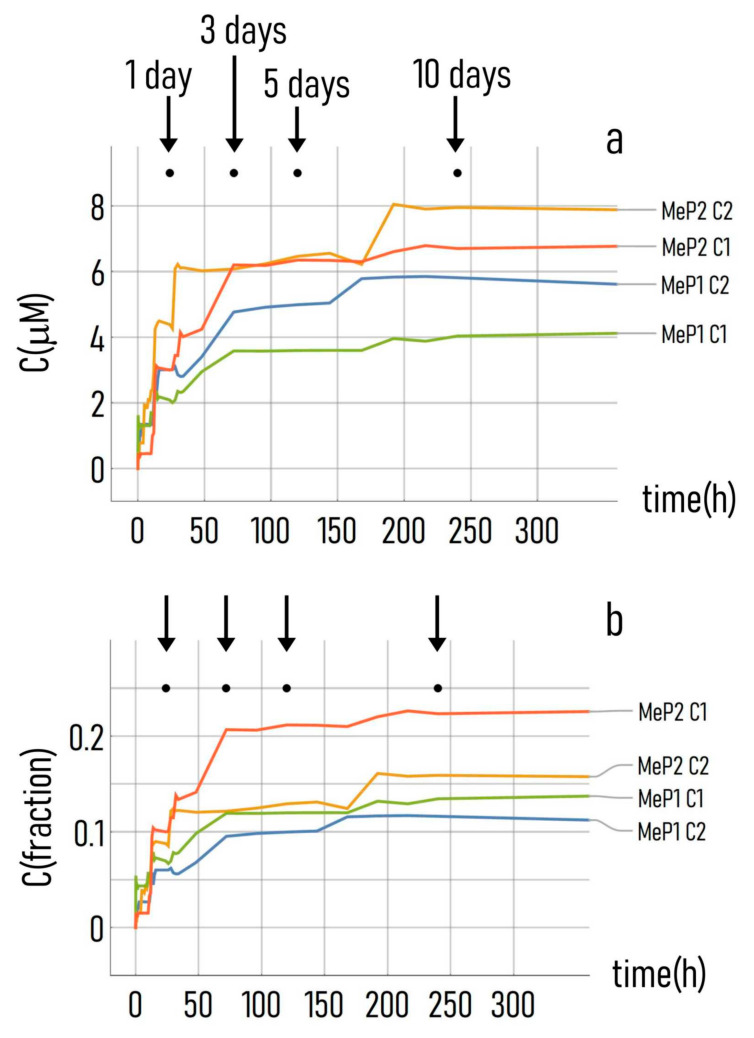
(**a**) Absolute and (**b**) normalized release profile of the anti-tumor drug $PtCl\left(O,O^{\prime }-acac\right)\left(DMSO\right)$ measured for ${\mathrm{MeP}}_{1}$ and ${\mathrm{MeP}}_{2}$ substrates up to 15 days from the beginning of the delivery.

**Figure 8 pharmaceutics-12-00481-f008:**
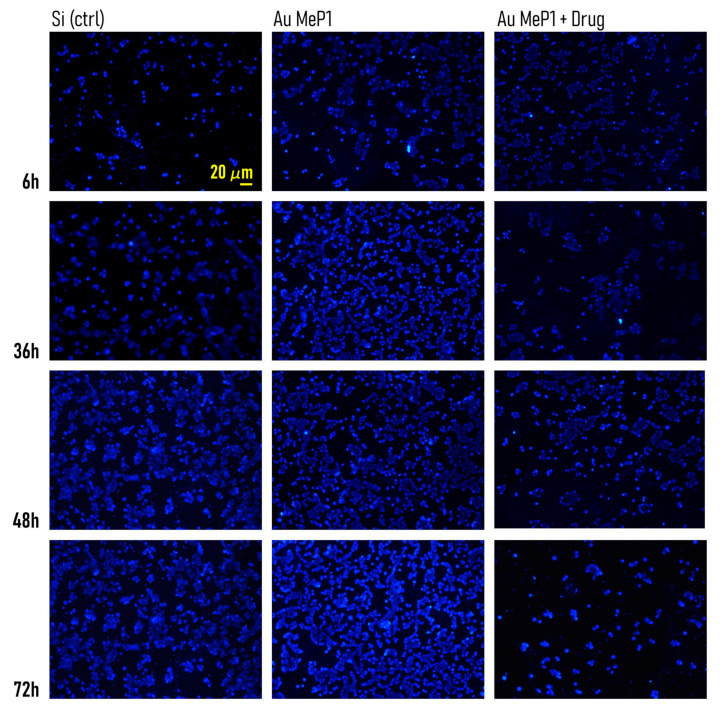
Fluorescence images of MCF-7 cancer cells growing on Au-MeP_1_ substrates with and without the delivery of the antitumor agent PtCl(O,O′-acac)(DMSO), at different time frames. In the experiments, unfunctionalized silicon was used as a control.

**Figure 9 pharmaceutics-12-00481-f009:**
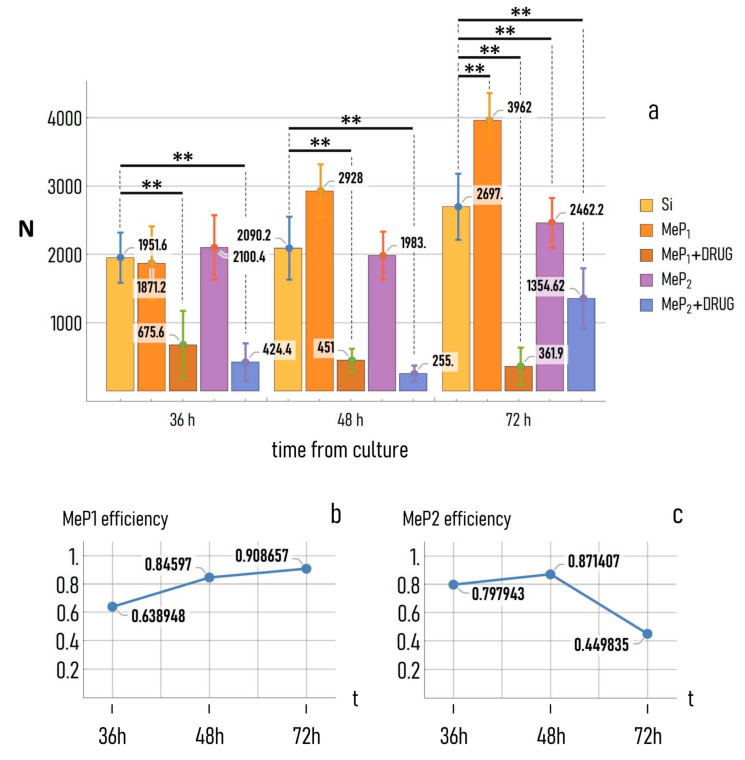
(**a**) Number of adhering cells on Au-MeP_1_ and Au-MeP_2_ substrates under and without the influence of the antitumor agent PtCl(O,O′-acac)(DMSO), at 36, 48, and 72 h from cell seeding, compared to plain silicon used as a control. All *p* values less than 0.05 are summarized with two asterisks. Efficacy of the antitumor drug on the cancer cells at different times from the initial release, for the (**b**) Au-MeP_1_ (**c**) and Au-MeP_2_ substrates.

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
