# Peer review of "Cell Theranostics on Mesoporous Silicon Substrates"

_pharmaceutics, 2020, doi:10.3390/pharmaceutics12050481_

Round 1

Reviewer 1 Report

This study is well designed, meticulously executed and also carefully presented. It offers important insights to using 3D scaffolds in tissue engineering and nanomedicine. Some minor points to address before it can be published. 

  1. It would be good to draw a schematic to show the step-by-step production of gold-functionalized mesoporous surfaces.
  2. What is the reason for choosing MCF-7 as the model cell?
  3. How do you envision the real-life application of this nanodevice? What are the possible hurdles that it needs to overcome?

Author Response

Manuscript ID: pharmaceutics-801604

Title: Cell theranostics on mesoporous silicon substrates

Journal: Pharmaceutics

Dear Editor,

we are pleased to submit to Pharmaceutics a revised version of the manuscript entitled Cell theranostics on mesoporous silicon substrates, and a detailed response to the comments of the Reviewers. The MS has been revised. Following suggestions from the Reviewers:

  1. We have written a new discussion section where we expound on the possible applications of the device and comment on the issues that should be addressed for improving the performance of the measurements.
  2. We have included in the article a schematics of the steps necessary to fabricate the mesoporous silicon substrates.
  3. We have commented on why we used MCF-7 cells to perform experiments.
  4. We have included details on the anti-cancer drug used in the study.
  5. We have discussed the effects of our nanomedicine formulation and of a drug-delivery approach on the half-life of drugs, especially anti-cancer drugs.
  6. We have rewritten parts of the MS.

Considering the corrections made to the manuscript and the enthusiastic comments of the Reviewers, we hope that the paper can be accepted for publication in this present form. We warmly thank the Reviewers that, with their comments, contributed to improve the work.

In the following, you will find the original comments from the reviewers (bold black text) and the point-by-point response of the authors (bold blue text). In the manuscript, new included portions are highlighted in YELLOW, whereas the text removed from the original version is highlighted in GRAY.

With Regards,

the Authors

Reviewer 2 Report

Coluccio et al in this manuscript has clearly provided a neat and clear work to demonstrate how nano-topography and delivery of therapeutics on cell growth and unbalance between these competing agents can accelerate the development of tumor cells. The manuscript is well written and the objectives addressed with clarity. 

The work will add value to the current research. the authors may have to indicate what drug they used.

Also, The authors use MCF-7 cells which are breast cancer cells with wild type p53 and very responsive. the authors may need to address how this concept would an aggressive cell line or type such as Triple-negative breast cancer. Does this system have an impact on normal cells? Least the authors must discuss them. Also, the caveats here that may need to overcome in future directions.

Will this help increase the half-life of a drug in the cells to be more efficacious? 

Author Response

(The authors gave the same response as above.)
